# Trust and Trade-Offs in Sharing Data for Precision Medicine: A National Survey of Singapore

**DOI:** 10.3390/jpm11090921

**Published:** 2021-09-16

**Authors:** Tamra Lysaght, Angela Ballantyne, Hui Jin Toh, Andrew Lau, Serene Ong, Owen Schaefer, Makoto Shiraishi, Willem van den Boom, Vicki Xafis, E Shyong Tai

**Affiliations:** 1Centre for Biomedical Ethics, Yong Loo Lin School of Medicine, National University of Singapore, Singapore 117597, Singapore; tlysaght@nus.edu.sg (T.L.); angela.ballantyne@nus.edu.sg (A.B.); serene.ong@u.nus.edu (S.O.); owen_schaefer@nus.edu.sg (O.S.); e0383648@u.nus.edu (M.S.); vicki.xafis@nus.edu.sg (V.X.); 2Department of Primary Health Care & General Practice, University of Otago, Wellington 6021, New Zealand; 3Projective Insights Consultants, Singapore 590003, Singapore; andrew_pc_lau@yahoo.com.sg; 4Yale-NUS College, National University of Singapore, Singapore 138527, Singapore; willem@yale-nus.edu.sg; 5Yong Loo Lin School of Medicine, National University of Singapore, Singapore 117597, Singapore; mdctes@nus.edu.sg; 6Saw Swee Hock School of Public Health, National University of Singapore, Singapore 117549, Singapore; 7Precision Health Research, Singapore 139234, Singapore

**Keywords:** Precision medicine, bioethics, trust, data sharing, survey, Singapore

## Abstract

Background: Precision medicine (PM) programs typically use broad consent. This approach requires maintenance of the social license and public trust. The ultimate success of PM programs will thus likely be contingent upon understanding public expectations about data sharing and establishing appropriate governance structures. There is a lack of data on public attitudes towards PM in Asia. Methods: The aim of the research was to measure the priorities and preferences of Singaporeans for sharing health-related data for PM. We used adaptive choice-based conjoint analysis (ACBC) with four attributes: uses, users, data sensitivity and consent. We recruited a representative sample of *n* = 1000 respondents for an in-person household survey. Results: Of the 1000 respondents, 52% were female and majority were in the age range of 40–59 years (40%), followed by 21–39 years (33%) and 60 years and above (27%). A total of 64% were generally willing to share de-identified health data for IRB-approved research without re-consent for each study. Government agencies and public institutions were the most trusted users of data. The importance of the four attributes on respondents’ willingness to share data were: users (39.5%), uses (28.5%), data sensitivity (19.5%), consent (12.6%). Most respondents found it acceptable for government agencies and hospitals to use de-identified data for health research with broad consent. Our sample was consistent with official government data on the target population with 52% being female and majority in the age range of 40–59 years (40%), followed by 21–39 years (33%) and 60 years and above (27%). Conclusions: While a significant body of prior research focuses on preferences for consent, our conjoint analysis found consent was the least important attribute for sharing data. Our findings suggest the social license for PM data sharing in Singapore currently supports linking health and genomic data, sharing with public institutions for health research and quality improvement; but does not support sharing with private health insurers or for private commercial use.

## 1. Introduction

Precision medicine (PM) broadly aims to provide more tailored care to patients based on genomic data analytics combined with other clinical, environmental and behavioral information [1,2]. The United States [3], United Kingdom [4], China [5] and several other countries are investing in PM programs with the goal to improve population health outcomes. PM programs typically seek broad consent from participants to store and share data with healthcare professionals and researchers at publicly funded institutions, as well as potentially with private enterprise and commercial partners. This model removes the need to obtain informed consent from participants each time the data are accessed for a specific purpose. However, it also relies on maintaining a social license for the program to operate without specific consent and with an implicit assumption that use of the data meets community expectations [6]. The ultimate success of PM programs will thus likely be contingent upon identifying public expectations about data sharing and management and establishing appropriate governance structures.

International literature on public preferences and attitudes towards health data sharing identifies numerous issues influencing the willingness of people to participate in PM, in particular privacy, risk minimization, data security, control, transparency, accountability, trust and the social value of the research [7]. The prior research suggests that, when given the option, people would prefer to have greater control over their health data [8,9,10,11,12,13,14]. However, other studies indicate that people may be willing to accept less control over their health data in exchange for public benefits [15]. In real life, decision-making requires trade-offs between competing values, and decision-based experiments have shown that respondents give greater weight to the potential benefits of health data sharing than the possible risk of privacy-related harms [16,17,18]. While in other research that considered trade-offs, participants gave priority to non-consensual uses of data provided that adequate privacy protections were in place and public benefits arising from the research were evident [19].

The existing empirical literature on public attitudes to genomics has been conducted primarily in Europe, North America and Australia, with a few studies done in Asia. The under-representation of Asian perspectives is problematic because it cannot be assumed that people everywhere perceive the issues and prioritize values in the same way. Some studies suggest that low participation rates of Asian minority populations in biobanks are due to cultural views about the sacredness of blood [20] and fear of discrimination [21]. As the parameters of any social license are likely to be culturally sensitive and context-specific, it is important to ensure diverse perspectives are included in the attitudinal research around genomics [22].

We report results of a nationally representative survey using adaptive choice-based conjoint analysis to measure the priorities and preferences that Singaporeans hold in relation to sharing health-related data for PM. Singapore is an ethnically diverse, city-island state in Southeast Asia with a high-income economy, robust health system, and national IT infrastructure for data storage and sharing [23,24]. In 2020, the Singapore government launched its ten-year National Precision Medicine (NPM) program [25]. Prior empirical studies have indicated that Singaporeans may be willing to participate in PM without the need for specific consent conditional upon robust data security systems, trustworthy governance, and public benefits [26,27]. However, it is unclear which attributes of data sharing are most important to Singaporeans. Understanding the relative importance of different data sharing arrangements will be crucial in establishing governance structures that reflect local priorities and values. Thus, our specific aims were to measure the relative value Singaporeans attribute to key data sharing arrangements, their willingness to share de-identified health data with broad consent, and their trust in various institutions using the data. We also aimed to determine what, if any, demographic features may influence these attitudes.

## 2. Methods

### 2.1. Survey Respondents

We recruited a stratified random sample of *n* = 1000 respondents of Singapore citizens and permanent residents for an in-person household survey. The data collection was outsourced to professional surveyors with knowledge and experience in conducting in-person household surveys in the multi-lingual and multi-ethnic context of Singapore (Projective Insights Consultants Pty Ltd, Singapore). The research team supervised the senior surveyor who conducted regular audits for data quality assurance. Households were identified from the Department of Statistics (DOS), Singapore Sampling Frame for 2019 [28] and randomly selected with stratified and quota sampling to ensure demographic representation. Singapore’s resident population is majority Chinese with minority Malay and Indian ethnic groups. To ensure adequate representation of the minority groups, Malay and Indian households were over-sampled by 3.0 and 7.5 percentage points, respectively, based on the DOS Sampling Frame (Appendix A). One respondent was chosen from the selected household and screened according to the eligibility criteria (i.e., population quota breakdown by age, sex and race) with household members deciding who would respond to the survey if more than one was eligible. Anyone from market research, public relations, the media, and pharmaceutical, medical or healthcare industry was also excluded to reduce bias. In the event that no household members were eligible or consented to the survey, or no one answered the door after three attempts, a replacement housing unit was sought to the right of the selected household (see Appendix A for further details).

### 2.2. Survey Instrument

The survey was developed based on findings from focus groups conducted as part of our mixed methods study [27,29] and an instrument designed for a conjoint analysis of public preferences for health data sharing arrangements in the United States [17]. Conjoint analysis estimates how people make complex decisions by balancing or making a trade-off between competing factors [30] and can be used to quantify numerous metrics simultaneously by presenting a series of side-by-side comparisons of scenarios with different attributes or features that respondents can choose from [31]. Different types of conjoint analyses are available, but we selected an adaptive choice-based conjoint (ACBC) analysis because it adapts and modifies the scenarios and options in response to input from the survey respondent. This approach has greater accuracy as it removes redundant options that the respondent will not select or does not choose as essential features of the preferred data sharing arrangement, and it more closely resembles the way choices are made in real-life than static survey questions.

The survey instrument was designed and administered using Lighthouse Studio from Sawtooth Pty Ltd, Utah, USA. [32], which specializes in software for ACBC analysis. The survey was divided into four sections: willingness to share data, trust in institutions, trade-offs in data sharing arrangements, and demographics.

Willingness to share data: This section comprised nominal scale questions asking about respondents’ willingness to share de-identified data under a model of broad consent and their reasons.

Trust in institutions: This section comprised 6-point Likert-scale questions (6—‘totally trust’ to 1—‘totally distrust’) asking respondents to rate how much they would trust a list of thirteen institutions/organizations to use their data. Two organizations (Ministry of Health and Facebook) were named specifically because these they have a high public profile and we were interested in the public response to a recognized organization.

Trade-offs in data sharing arrangements: This section comprised: (i) preliminary questions to adapt scenarios presented to each respondent, and then (ii) three different scenarios (each with four attributes) were presented side-by-side repeatedly for the ACBC analysis. We used the four attributes uses, users, data sensitivity and consent with the following options:Uses—the purposes respondents are willing to share data for.
Health research;Quality improvement;Commercial uses.
Users—with whom respondents are willing to share data.Government agencies;Private insurers;Pharmaceutical/biotech companies;Universities/research institutes;Hospitals.
Data sensitivity—the types of data respondents would be sharing.Identifiable genetic data and your medical history;Identifiable genetic data without your medical history;De-identified genetic data and your de-identified medical history;De-identified genetic data without your medical history.
Consent—the degree of control over the data.Consent for every study;One time consent only for all studies;Consent for some specific studies.


Demographics: This section asked for standard demographic information to evaluate the representativeness of the sample in terms of age, gender and ethnicity, to test for relationships between these characteristics and survey responses, as well as level of education, household type and income, religious identity, and respondents self-assessed health status.

The instrument was pilot tested with 20 respondents before being launched and refined to minimize the duration and respondent fatigue. Prior to starting the survey, the interviewers showed respondents a short 2–3 min video demonstrating the concept of PM [33] and flash cards with definitions of key terms (see Appendix A). For ethnic Chinese and Malay households, the survey and supporting information was translated into Mandarin and Malay for respondents who preferred to respond in their mother tongue. Interviews took an average of seven minutes and respondents were given a SGD 10 (~USD 7.50) shopping voucher as a token of appreciation for completing the survey. The research team supervised the lead survey administrator who conducted regular audits to verify data quality.

### 2.3. Data Analysis

Confidence intervals for the institutional trust scores were computed based on *t* statistics. Additionally, we considered Spearman’s rank correlation coefficients between scores of any pair of institutions. The ACBC responses were analyzed in Lighthouse Studio [32] using hierarchical Bayes with weighting of respondents to account for the over-sampling of Malay and Indian households according to the Singapore government demographic statistics (i.e., 76.0% Chinese, 15.0% Indian, 7.5% Malay and 1.5% Others) [34]. Conjoint analysis estimates a numerical score, called utility, for each scenario. A scenario with a higher utility is more likely to be preferred over one with a lower utility. Additionally, ACBC estimates a utility threshold above which most respondents are willing to share their data. Specifically, the analysis estimates part-worth utilities for each level of each attribute. The total utility of a scenario then derives from these part-worth utilities. Importance, which is a score of how influential an attribute is for the respondents’ choices, is derived from the part-worth utilities. In addition to performing ACBC analysis on all responses, we ran separate analyses on various demographic subgroups. Below, we discuss choice simulations that use the method “randomized first choice” and were based on the ACBC results.

### 2.4. Ethics Oversight

The National University of Singapore Institutional Review Board granted an exemption from ethical review (S-20-145E). All participants gave their verbal consent and received an information sheet explaining the aims, benefits and risks of the study.

## 3. Results

Twenty-one interviewers surveyed the final sample size of *n* = 1000 once they had disbursed all the shopping vouchers. Fifty-five (55) surveys were partially completed and excluded from the analysis. The number of completed surveys over the number of those started (1055) resulted in a high compliance rate of 95%, which was likely bolstered by the relatively short duration of the survey (average 7 min), the reputation of the authorization institution, and the shopping voucher. Interviewers did not systematically record the number of households approached from the sampling frame and the number that had an ineligible household member to complete the survey. Based on estimates for these numbers, the response rate was between 32% and 43% (Appendix A).

### 3.1. Population Demographics and Willingness to Share Data

Table 1 summarizes demographic and socio-economic characteristics, and Figure 1 summarizes the reasons on willingness to share data. The demographics and socio-economic variables generally reflect the characteristics of the resident population of Singapore (i.e., citizens and permanent residents) between the ages 21 years and 85 notwithstanding the over-sampling of Malays and Indians. Per Figure 1, slightly less than two-thirds (64%) were willing to share de-identified health data with institutions for IRB-approved research without consent for each study. Among those, 67% selected the option ‘potential benefits of health research are greater than the potential risks’, and 19% chose ‘it would be too troublesome to consent to every study’ as one of their reasons. Of the 36% who were unwilling to share data without giving consent for each study, 60% chose the option ‘to have control at all times’, and 35% opted for ‘don’t trust the research processes and protections’.

### 3.2. Trust in Institutions

In Figure 2, respondents indicated higher levels of trust in public institutions, especially in the Singapore government, than the private sector. Appendix A contains Spearman’s correlation coefficients between trust scores. Scores among public institutions were either moderately or strongly correlated (correlation ranging from 0.44 to 0.93). Conversely, correlations between public and private institutions were weaker, ranging from 0.11 to 0.47, except for a correlation of 0.53 between pharmacies and pharmaceutical companies. Overseas universities had weak to moderate correlation with public institutions, but a higher correlation with private entities such as Facebook (*R* = 0.50), private insurers (*R* = 0.77), pharmaceutical companies (*R* = 0.77), and private genetic testing companies (*R* = 0.79).

### 3.3. Trade-Offs in Data Sharing Arrangements

As per ACBC results shown in Figure 3, the users attribute was the most influential deciding factor in their choice to share data. The uses attribute was the second most influential, while data sensitivity, and then consent were the least influential. The part-worth utilities for the users reflect their trust scores, with the highest utility for government agencies. Relatedly for the attribute uses of the data, respondents were least willing to share data for commercial purposes. Respondents were more willing to share data in scenarios which included medical history in “data sensitivity” than those that did not.

The Part-worth utilities of the attribute levels and their importance from the ACBC analysis. A higher utility corresponds with increased willingness to share data. The width of the boxes corresponds with 95% confidence intervals.

The part-worth utilities and the acceptance threshold derived from ACBC can be summarized via simulation of what proportion of respondents would deem a specific scenario acceptable. Appendix A contains simulation results with 95% confidence intervals indicating statistical significance. They reveal that most respondents found it acceptable to share de-identified data for government agencies and hospitals for the purpose of health research irrespective of the consent model (ranging between 76 and 88% for government and 71 and 85% for hospitals depending on consent type and data sensitivity). This level of acceptance was significantly greater than universities/research institutes (48–66%), pharma/biotech companies (31–49%) and insurers (14–27%). Acceptability for all users was also significantly less when sharing data for quality improvement than health research (62–76% for government; 52–74% for hospitals; 33–51% for universities/research institutes; 20–35% for pharma/biotech; and 9–18% for insurers). Commercial/private uses were widely held as unacceptable for all users with responses ranging from 62% for government to 98% for insurers. Respondents did not distinguish greatly between consent for every study versus some studies, but one-time consent was significantly less acceptable across all scenarios.

Finally, we conducted subgroup analyses. Appendix A suggest that certain demographic variables may be associated with preferences for certain levels of attributes over others. However, when we analyzed the choice-based simulations for these subgroups with 95% confidence intervals, many of the apparent differences were statistically insignificant. As shown in Appendix A, we found statistically significant differences between 21 and 39 years and 60 and above year age groups with younger respondents being more likely to share de-identified data for the purpose of health research when the users were universities, pharma/biotech companies and private insurers. Higher income earners (>SGD 10 k/month) were statistically more likely to share de-identified data for health research with universities/research institutes and pharma/biotech companies than the lowest income bracket (<SGD 2999 k/month), and tertiary educated respondents were also more likely to share with these users than those with nil to secondary school education. Of note, no statistically significant differences were found between religious groups, ethnicity or gender in any of the scenarios we tested.

## 4. Discussion

This survey measured public attitudes in Singapore towards the sharing of health-related data. Specifically, it measured the relative value of key data sharing arrangements for Singaporeans, their levels of trust in public versus private institutions, and their willingness to share de-identified data under models of broad consent.

While a large body of prior research focuses on preferences for consent, our conjoint analysis found that consent was the least important attribute for sharing data when considered against the kind of users and uses of the data. This finding is important for interpreting the prior literature on consent preferences for data sharing. Public attitudes vary greatly on the need for consent [12,35], with some people expecting to consent for each data use [9], some willing to provide one time (broad) consent [36,37,38] and some happy for data to be used without consent [19,39]: two thirds of the participants in our study indicated that they were willing to share de-identified health data with institutions for IRB-approved research without specific consent. It is tempting to attribute the apparent differences between the relative weight given to consent in this study compared with others in Northern America and Europe to cultural norms in Asia that are assumed to place less emphasis on individual liberty and consent. However, such assumptions are contestable and not well-supported empirically. The differences with prior studies may instead reflect underlying normative assumptions that focus research questions too narrowly on binary preferences and inadvertently over-emphasize the importance of consent, especially when other (potentially more important) considerations are taken into account.

A recent systematic literature review reported that, when given the choice, individuals often want more control and prefer specific consent [8]. However, consent models that prioritize autonomy and individual control over data use, for example specific and dynamic consent, are expensive and resource intensive—for both the data holders and the data subjects [40,41]. Our findings suggest that the users and uses of the data will likely influence public trust and willingness to share data more than either the sensitivity of the data or the consent procedures. Prior research has also consistently demonstrated the importance of public benefits for acceptability of data-linkage research [19,42,43]. Thus, it is perhaps more important to ensure data users and uses are transparent, and that governance processes and access control align with the priorities of the population rather than persisting with complex and costly consent procedures that do not feature prominently in peoples’ concerns.

International studies have found that the types of data being linked was an important factor in publics’ willingness to share data with the type of data being linked roughly twice as important as who the researchers are [44], and that participants preferred less data linkages [45]. However, our study found that data sensitivity (or type of data being linked) was of less importance (19.5%) than either the users (39.5%) or the uses (28.5%), and participants preferred data sharing arrangements that linked genome data with their electronic medical history. This result is interesting because linking genomic data with medical history increases the amount of data shared, thereby increasing the potential for re-identification and the potential magnitude of harm due to a data breach. However, it may simply be that Singaporeans are cognizant of the greater research value derived from linking medical history data with genomic data than the value derived from the use of unlinked data sets.

This finding may have also have arisen because the short video respondents viewed prior to the survey explained the concept of linking genome data with electronic medicals records and the purposes for which they may be shared. However, rather than being a concern for bias, it may demonstrate how clear and comprehensibly presented information can positively impact people’s understanding of complex scientific concepts and process, as has been demonstrated elsewhere [19,26]. Moreover, prior studies that have found a preference for linking less data, and that data sensitivity/type was an important attribute, compared with linking different types of health data to cross-sectorial related data (e.g., health data linked with education, employment or shopping records) [44,45] to find that respondents objected to this type of cross-sectorial data linkage. Whereas our study only looked at preferences for linking different types of health data (genome sequencing and medical histories). Thus, these results do not necessarily conflict.

Prior research has shown general public reluctance to share data with commercial companies [8,9,46], primarily due to concerns about exploitation/profiteering [39,47] and a lack of familiarity [48,49] with intended uses of the data by commercial companies. However, there is limited research that differentiates between different commercial entities with an interest in health data. Research in Great Britain found participants in workshops objected to all access to health data by insurance companies because it was seen to be detrimental to individual interests, as well as societal interests, as it works against the principle of a public health service [48]. Our findings demonstrate that Singaporeans were significantly less willing to share data with private insurance than with pharmaceutical companies. This finding is consistent with prior qualitative studies in Singapore that have reported an openness to sharing with pharmaceutical companies for research that can generate public benefit, but not private insurers [27]. It is also consistent with the relative trust of users in our survey, which measured significantly lower levels for private insurers.

Trust relies on motivations and competence [50,51]. Prior research suggests sectors with the highest trust for data sharing are healthcare professionals, health services and public sector researchers, whereas government agencies are less or not trusted, and commercial entities are trusted less [44,52,53]. However, in our study, the organizations with the highest trust and utility were government agencies and, in particular, the Ministry of Health in Singapore. A recent study in the Republic of Korea also found a strong preference for PM research to be carried out by government agencies [54] suggesting that East and Southeast Asian ‘developmental states’ [55] may exhibit higher trust in governments, including with respect to data research, than countries in Europe and North America. Singapore, in general, has relatively high levels of trust in government [56,57] and prior qualitative research in Singapore has also reported high trust in government agencies to manage and use PM data [27]. Furthermore, it is possible that Singapore’s management of the SARS-CoV-2 pandemic, widely recognized as effective and competent [58,59], boosted Singaporeans’ trust in government at the time when our survey was conducted; indeed, according to the Elderman Trust Barometer survey [60], 2020 saw exceptionally high levels of trust in governments globally. Thus, our findings appear to be consistent with this trend.

While these survey results could be motivated by a variety of considerations, they are in line with a normative framework that prioritizes data governance in the public interest rather than promoting individual control over data or maximally restricting sensitive data. Such a framework has over the past decade been applied to biobanking governance [61,62,63,64], and its principles are directly applicable to data governance as well [65]. On this model, the primary purpose of data governance is to ensure data are used responsibly and in ways that are aligned with the priorities and interests of a given society. Individual consent for particular studies is less crucial when other aspects of governance are well-maintained, and data sensitivity is less concerning when data are being used by trustworthy institutions. Public institutions like government agencies or publicly funded universities are in turn generally considered to be more trustworthy than private enterprise because they are constituted for the public benefit, rather than benefits of shareholders or owners.

Finally, the subgroup analysis revealed statistically significant differences between the lowest and highest age brackets, income earners and education levels with some of the user and use attributes under various consent arrangements. However, it is difficult to interpret the practical significance of these differences or ascertain the reasons why without further inquiry. For example, although the tertiary educated and higher income earners were statistically more likely to share data with private insurers, it does not mean they actually would, given that the overall willingness of this sub-group was strikingly less than other users. Furthermore, even if they were willing to do so, the reasons why education and income might influence someone’s decision to share data with private insurers is unclear: it may be that people with tertiary education and/or higher incomes are more likely to purchase private insurance and are thus more familiar with insurance providers and see their value in the wider healthcare system.

Of greater interest is the lack of statistically significant differences across the demographic groups, particularly with gender, ethnicity and religion. Even where differences were statistically significant, the relative importance of the various attributes was much larger than the differences between groups, indicating an unexpected homogeneity in attitudes across the population. The finding may be surprising given the ethnic and cultural diversity of Singapore and policymakers may be reassured in developing public communications strategies that are more homogenous than if significant associations were measured between these demographic variables and acceptable data sharing arrangements.

Readers should note the inherent limitations of quantitative surveys when interpreting these findings. As with any survey instrument, ours was designed to measure the breadth of a large representative sample population responding to a limited set of options and, although the ACBC design can more confidently indicate choices that are closer to the way people make decisions in real life, such an instrument cannot also investigate or capture the justifications for those choices. Therefore, further research that can go into greater depth to understand how Singaporeans prioritize certain data sharing arrangements and why particular groups might preference one attribute over another (or not) is needed.

Surveys are also exposed to self-selection bias whereby people who were more supportive of science in general may be more likely to choose to respond to this survey [66], which asked about a hypothetical program of scientific research. To mitigate this bias, we excluded households with potential conflicting professional and financial interests in PM and conducting pilot testing to ensure the survey duration was not so onerous that only highly motivated respondents would complete the survey. However, we did not question why eligible household members refused consent and members of the household were able to decide who responded to the survey, whereby the most interested individual may have self-selected. However, given the context of the global coronavirus pandemic and the social distancing requirements in place in Singapore at the time of data collection, we believe personal risk thresholds for interacting with strangers may have had a greater impact on individuals’ decision to participate, than pro-science selection biases.

Our estimated response rate was reasonably high (32–43%) and we had a high completion rate, which does not indicate selection bias. Additionally, our results showing willingness to participate in PM are broadly consistent with international survey findings [67,68,69], suggesting that any potential self-selection bias did not unduly distort the findings. In terms of any impact of self-selection bias on the practical application of the research findings, we note that a successful PM or bio-banking program only requires participation from a subset of the community and that is likely to be those who are more pro-science. There are future research opportunities for comparing survey results of hypothetical data sharing propositions with the recruitment rates of actual research programs that share linked genome and health data, both from the general population and in clinical settings with disease-specific populations.

## 5. Conclusions

In conclusion, the results from this study suggest that there is broad support in Singapore for linking health and genomic datasets and sharing data with public institutions for health research and quality improvement. At this stage, there is little support for sharing data with private health insurers or for private commercial use. Willingness to share data depends more on the users and the uses to which the data was put than the sensitivity or the consent model. While a significant body of prior research focuses on preferences for consent, our conjoint analysis found it was the least important attribute for sharing data. Investing in expensive and resource intensive consent models, such as specific consent or dynamic consent, may therefore have minimal impact on public acceptance of data sharing in Singapore. There were surprisingly few differences between subgroups in this study, suggesting reasonably homogenous attitudes towards data sharing within Singapore. Government agencies and public institutions are the most trusted users of data. In order to demonstrate trustworthiness and ensure ongoing support for data sharing in Singapore, those responsible for governing data may wish to consider implementing processes that ensures data are used for projects with public benefit and this is transparent.

## Figures and Tables

**Figure 1 jpm-11-00921-f001:**
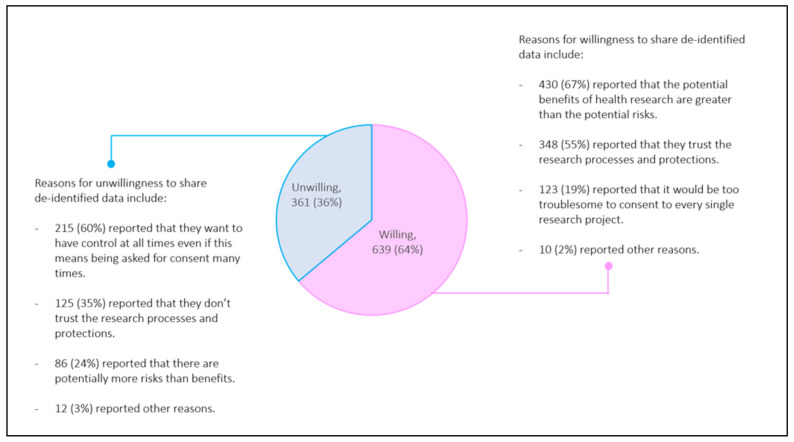
Willingness to share de-identified data for Institutional Review Board approved health research without needing consent for each study.

**Figure 2 jpm-11-00921-f002:**
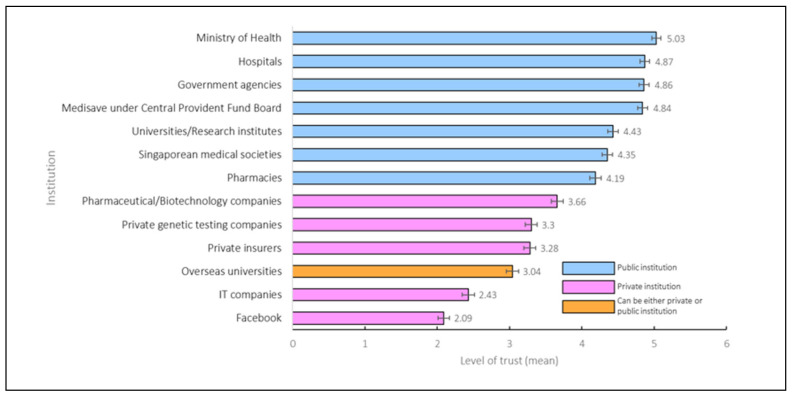
Mean trust scores for institutions using de-identified health data for Institutional Review Board approved research. A score of 6 represents “trust totally” and 1 represents “distrust totally”. The whiskers indicate 95% confidence intervals.

**Figure 3 jpm-11-00921-f003:**
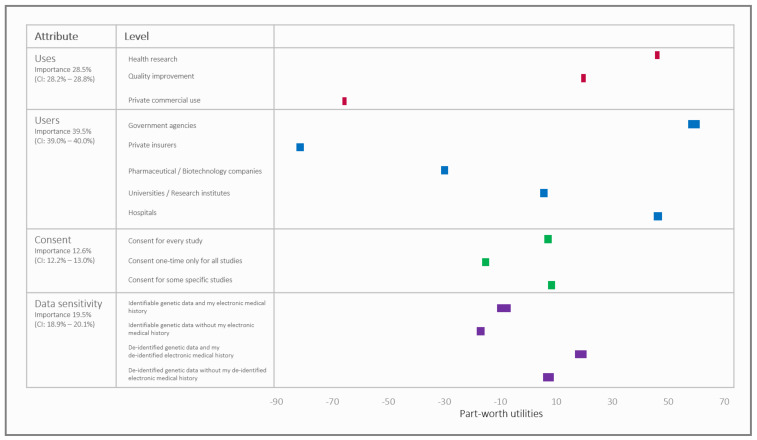
Part-worth utilities of the attribute levels and their importance from the ACBC analysis. A higher utility corresponds with increased willingness to share data. The width of the boxes corresponds with 95% confidence intervals.

**Table 1 jpm-11-00921-t001:** Respondent characteristics (sample size *n* = 1000).

Characteristic	No. of Respondents
Age bracket	
21–29	149
30–39	188
40–49	199
50–59	195
60–69	162
70–79	91
≥80	16
Gender	
Female	519
Male	481
Ethnicity	
Chinese	635
Malay	178
Indian	177
Others	10
Religion	
Buddhism	274
Christianity	155
Hinduism	111
Islam	219
Sikhism	41
Taoism	20
No religion	174
Others	6
Self-rated health	
Poor	11
Fair	110
Good	471
Very good	320
Excellent	88
Monthly household income, SGD	
No income	78
≤2999	216
3000–5999	195
6000–9999	156
10,000–14,999	80
≥15,000	30
No response	245
Housing Type	
1-room flat	30
2-room flat	53
3-room flat	264
4-room flat	374
5-room flat	211
Condominium	33
Landed property	35
Educational level	
No formal education	40
Primary	139
Secondary/Post-secondary	291
A-Level/Polytechnic	166
Tertiary education	364

## Data Availability

The datasets generated and/or analyzed during the current study will be uploaded into NUScholarBank for public access at the completion of the study, which ends in October 2021.

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
