# Peer review of "Trust and Trade-Offs in Sharing Data for Precision Medicine: A National Survey of Singapore"

_jpm, 2021, doi:10.3390/jpm11090921_

Round 1
Reviewer 1 Report
Lysaght and coworkers investigated the priorities and preferences of Singaporeans for sharing health-related data for personalized medicine programs. 64% of respondents were generally willing to share de-identified health data from previous IRB approved studies; respondents’ willingness to share data depended by: users (39.5%), uses (28.5%), data sensitivity (19.5%), consent (12.6%).
While presenting promising data on the willingness of the general population to share health related data, this study has two important flaws, in my opinion:
- individuals participating in this survey agreed to participate in scientific research - i.e. being "science-friendly"- , and in turn, they resulted to be more open to share health data. This could represent a selection bias. How many individuals were contacted before recruiting the 1000 participating in the study? How many rejected the participation and why? Probably those are the people not willing to share their health data too. Are they the majority?
- Only 11 individuals in the cohort report a poor self-rated health. The majority of the interviewed individuals do not have health problems and are open to share their health-related data. Would sick individuals - especially those affected by genetic disorders or transmissible diseases - be less willing to share their data? The same survey run in hospitalized individuals may help to answer this question and would be of value for the presented study.
Other minor concerns:
a. what is the sex and age of the 21 investigators that run the interviews?
b. does any results change accordingly to the gender of the interviewer?
Author Response
Lysaght and coworkers investigated the priorities and preferences of Singaporeans for sharing health-related data for personalized medicine programs. 64% of respondents were generally willing to share de-identified health data from previous IRB approved studies; respondents’ willingness to share data depended by: users (39.5%), uses (28.5%), data sensitivity (19.5%), consent (12.6%).
Authors' response to comment #1:
Thank you for your kind comments.
While presenting promising data on the willingness of the general population to share health related data, this study has two important flaws, in my opinion:
- individuals participating in this survey agreed to participate in scientific research - i.e. being "science-friendly"-, and in turn, they resulted to be more open to share health data. This could represent a selection bias. How many individuals were contacted before recruiting the 1000 participating in the study? How many rejected the participation and why? Probably those are the people not willing to share their health data too. Are they the majority?
Authors' response to comment #2:
Thank you for inquiring about the response rate and whether a selection bias of science-friendly people might have influenced the survey responses. We are unaware of evidence supporting the characterisation of survey respondents in such a way and did not include screening questions to identify support for scientific research more generally. As described in the methods on page 4, we excluded household members from relevant industries (i.e. pharmaceutical, medical or healthcare industry) to reduce the risk of bias from those who with potentially conflicting professional and/or financial interests in PM. From the estimates shown in Supplementary Table 2, approximately 15% of eligible household members invited to participate refused consent. Although interviewers did not survey reasons for refusal, this figure is not indicative of a larger majority of people not willing to share. We note this caveat and need for further research into reasons for refusals in the discussion on page 17.
Reviewer's comment #3:
- Only 11 individuals in the cohort report a poor self-rated health. The majority of the interviewed individuals do not have health problems and are open to share their health-related data. Would sick individuals - especially those affected by genetic disorders or transmissible diseases - be less willing to share their data? The same survey run in hospitalized individuals may help to answer this question and would be of value for the presented study.
Authors' response to comment #3:
Thank you for this observation. As there were only 11 with poor health, the number with fair health was larger, so we combine those who reported poor health and those with fair health into one group for the sub-group analysis. We did not find any statistical difference in willingness to share de-identified health-related data between those who reported poor/fair health and those with good/very good/excellent health. However, we would be cautious about interpreting that result as the item is self-reported and not validated against the actual health status of individuals in the population. Respondents may be unaware of underlying health conditions and/or simply do not wish to report being in poor health, especially if other household members were present during the interview.
Administering the survey to hospitalised patients would be an excellent follow up study but it is out of scope of the current paper since PM aims to recruit healthy volunteers from the general population rather than patients with specific diseases in hospital settings. Furthermore, prior empirical research in Singapore (see Bylstra et al 2017) and overseas (Brall et al 2021, Lemke et al 2010, Whiddett et al 2016) suggest that people with genetic diseases are more willing to participate in research with data sharing arrangements than not because they see themselves or ‘future others’ like them gaining the benefits. Thus, we find it unlikely that the self-reported health of respondents has positively biased the results but have added the possibility for future research to examine this question on page 17.
Reviewer's comment #4:
Other minor concerns:
- what is the sex and age of the 21 investigators that run the interviews?
- does any results change accordingly to the gender of the interviewer?
Authors' response to comment #4:
There were 11 (52%) female and 10 (48%) male interviewers. Four (19%) interviewers were in the age group 31-39 years, 6 (29%) in the age group 41-49 years, 9 (43%) in the age group 51-59 years, and 2 (10%) in the age group 61-69 years.
The 11 female interviewers interviewed 518 people while the 10 male interviewers interviewed the rest (482 people). We ran choice simulation analyses to look for significant differences in any of the scenarios between those who were interviewed by a male interviewer and those who were interviewed by a female interviewer, but did not find any. We have added results of these analyses to Supplementary Table 5.
Reviewer 2 Report
In this paper, the authors raise the problem of data sharing data for precision medicine. This work is well-written and easy to follow. The authors explain their motivation well, explain the problem and challenges. The survey shows the data on the willingness of data sharing, trust scores for institutions. Finally, obtained data are analyzed and discussed in detail.
Overall, I think this paper fits well for publication in Journal of Personalized Medicine.
Author Response
Reviewer's comments:
In this paper, the authors raise the problem of data sharing data for precision medicine. This work is well-written and easy to follow. The authors explain their motivation well, explain the problem and challenges. The survey shows the data on the willingness of data sharing, trust scores for institutions. Finally, obtained data are analysed and discussed in detail. Overall, I think this paper fits well for publication in Journal of Personalized Medicine.
Authors' response:
Thank you for your kind comments.
This manuscript is a resubmission of an earlier submission. The following is a list of the peer review reports and author responses from that submission.
Round 1
Reviewer 1 Report
In this paper, the authors raise the problem of data sharing data for precision medicine. This work is well-written and easy to follow. The authors explain their motivation well, explain the problem and challenges. The survey shows the data on the willingness of data sharing, trust scores for institutions. Finally, obtained data are analyzed and discussed in detail.
Overall, I think this paper fits well for publication in Journal of Personalized Medicine.
Author Response
Reviewer's comments:
In this paper, the authors raise the problem of data sharing data for precision medicine. This work is well-written and easy to follow. The authors explain their motivation well, explain the problem and challenges. The survey shows the data on the willingness of data sharing, trust scores for institutions. Finally, obtained data are analysed and discussed in detail. Overall, I think this paper fits well for publication in Journal of Personalized Medicine.
Authors' response:
Thank you for your kind comments.
Reviewer 2 Report
Lysaght and coworkers investigated the priorities and preferences of Singaporeans for sharing health-related data for personalized medicine programs. 64% of respondents were generally willing to share de-identified health data from previous IRB approved studies; respondents’ willingness to share data depended by: users (39.5%), uses (28.5%), data sensitivity (19.5%), consent (12.6%).
While presenting promising data on the willingness of the general population to share health related data, this study has two important flaws, in my opinion:
- individuals participating in this survey agreed to participate in scientific research - i.e. being "science-friendly"- , and in turn, they resulted to be more open to share health data. This could represent a selection bias. How many individuals were contacted before recruiting the 1000 participating in the study? How many rejected the participation and why? Probably those are the people not willing to share their health data too. Are they the majority?
- Only 11 individuals in the cohort report a poor self-rated health. The majority of the interviewed individuals do not have health problems and are open to share their health-related data. Would sick individuals - especially those affected by genetic disorders or transmissible diseases - be less willing to share their data? The same survey run in hospitalized individuals may help to answer this question and would be of value for the presented study.
Other minor concerns:
a. what is the sex and age of the 21 investigators that run the interviews?
b. does any results change accordingly to the gender of the interviewer?
Author Response

(The authors gave the same response as above.)

Round 2
Reviewer 2 Report
None
Author Response
Thank you